# A Multi-Center Prospective Study on Post-Vaccination Humoral Response to SARS-CoV-2 in Polish Long-Term Care Facility Residents: Associations with COVID-19 Clinical Course and Comorbidities

**DOI:** 10.3390/idr17040089

**Published:** 2025-07-24

**Authors:** Justyna Brodowicz, Piotr Heczko, Estera Jachowicz-Matczak, Mateusz Gajda, Katarzyna Gawlik, Dorota Pawlica-Gosiewska, Bogdan Solnica, Jadwiga Wójkowska-Mach

**Affiliations:** 1Department of Clinical Biochemistry, Jagiellonian University Medical College, 31-121 Krakow, Poland; justyna.brodowicz@uj.edu.pl (J.B.); k.gawlik@uj.edu.pl (K.G.); dorota.pawlica@uj.edu.pl (D.P.-G.); bogdan.solnica@uj.edu.pl (B.S.); 2Department of Microbiology, Faculty of Medicine, Jagiellonian University Medical College, 31-121 Krakow, Poland; mbheczko@cyf-kr.edu.pl; 3Department of Infection Control and Mycology, Faculty of Medicine, Jagiellonian University Medical College, 31-121 Krakow, Poland; estera.jachowicz-matczak@uj.edu.pl (E.J.-M.); mateusz14.gajda@uj.edu.pl (M.G.)

**Keywords:** long-term-care facilities, COVID-19, serological assays, ELISA, SARS-CoV-2 antibodies, vaccination coverage, public health, immunological response

## Abstract

Background: Vaccination effectively reduces the risk of infection, including COVID-19 yet older adults often receive insufficient attention despite their increased vulnerability. The study aimed to correlate serological results with underlying conditions, vaccination status, and COVID-19 history. Methods: This non-interventional, multicenter study aimed to assess vaccination coverage and SARS-CoV-2 antibody levels among residents of eight long-term care facilities (LTCFs) in Southern Poland. Data collection took place between January and June 2022, with 429 participants recruited based on their ability to provide informed consent and their residency in LTCFs. Sociodemographic data, medical history, and COVID-19-related information—including infection history and vaccination status—were collected through surveys. Blood samples were obtained for serological testing using enzyme-linked immunosorbent assays (ELISA) to detect anti-SARS-CoV-2 antibodies. Statistical analysis, including Spearman’s correlation, revealed significant associations between antibody levels and vaccination status, as well as between RT-PCR-confirmed COVID-19 infections and higher antibody titers. Results: Among the seven different qualitative serological, only the Anti-SARS-CoV-2 NCP (IgG) and Anti-SARS-CoV-2 (IgA) tests showed a positive correlation with the Anti-SARS-CoV-2 QuantiVac (IgG) test, which was used as a comparator. A weak correlation was noted with the age of the residents. Conclusions: Our findings suggest that vaccination positively influences antibody responses, underscoring the importance of immunization among LTCF residents. Additionally, certain comorbidities—such as degenerative joint disease and diabetes—showed weak correlations with higher antibody levels. This study provides valuable insights into the humoral immune response to COVID-19 in vulnerable populations residing in LTCFs.

## 1. Introduction

Residents of long-term care facilities (LTCFs) represent a population particularly vulnerable to infections due to advanced age, multiple comorbidities, and compromised immune competence. Additionally, their physical limitations often necessitate frequent assistance and support. At the same time, the communal living environment in LTCFs results in more uniform exposure to pathogens, including coronaviruses, compared to the general population. From an epidemiological perspective, the risk of acquiring infection is highly homogeneous, and the vulnerability due to age and underlying health conditions further increases susceptibility [1].

This characteristic became especially relevant during the COVID-19 pandemic, as LTCF residents were at increased risk of severe disease. In Poland, in 2020, there were 3384 confirmed COVID-19 cases and 75 deaths per 100,000 inhabitants [2,3,4].

The epidemiology of COVID-19 changed dramatically following the development and licencing of vaccines against SARS-CoV-2 and the subsequent launch of vaccination campaigns [5]. By 31 December 2022, a total of 268,677,438 vaccine doses had been administered across Europe [6]. By the end of 2021, nearly 56% of the Polish population had received at least one dose of the vaccine. Following the introduction of vaccines, numerous studies were published monitoring specific antibody levels in vaccinated populations [7,8,9].

SARS-CoV-2, the causative agent of COVID-19, is a positive-sense single-stranded RNA virus encoding a large polyprotein (1a and 1ab), four structural proteins—spike (S), envelope (E), membrane (M), and nucleocapsid (NC)—and several accessory proteins [10]. The spike (S) protein plays a crucial role in viral entry and is divided into two subunits: S1, responsible for cell attachment, and S2, responsible for membrane fusion [11]. Angiotensin-converting enzyme 2 (ACE2) was identified as the cellular receptor for SARS-CoV [12]. A fragment of the S1 subunit (residues 318–520) constitutes the receptor-binding domain (RBD), which mediates ACE2 binding and viral entry [13]. The RBD contains major neutralizing epitopes, making it a key target for the human immune response [11].

The homotrimeric spike glycoprotein is the primary target for neutralizing antibodies. It is a class I fusion glycoprotein that undergoes proteolytic cleavage into S1 and S2 subunits. The S2 subunit includes fusion peptide, transmembrane domain, and heptad repeat sequences essential for membrane fusion [14]. The S1 subunit contains the RBD, which mediates viral attachment to the host receptor dipeptidyl peptidase 4 (DPP4) [15].

Currently available commercial serological assays—including ELISAs, CLIAs, and LFIAs—are designed to detect SARS-CoV-2 antibodies. However, their performance varies depending on the technology used, timing of sample collection, and reference methods. Clinical validation is ongoing, but recent meta-analyses and systematic reviews have highlighted significant heterogeneity and potential bias [16].

Both IgM and IgG ELISA-based tests demonstrate over 95% specificity for COVID-19 diagnosis. Most antibodies are produced against the nucleocapsid (NC) protein, making anti-NC tests highly sensitive. In contrast, antibodies targeting the RBD of the spike protein are more specific. Therefore, using both antigens to detect IgG and IgM enhances sensitivity [17].

Recently, quantitative assays detecting IgG antibodies against the S1 protein have been developed. Their performance shows strong correlation with virus neutralization tests, suggesting their utility in confirming and monitoring recent and past SARS-CoV-2 infections [18,19].

While the sensitivity of serological tests varies, their specificity remains generally high. Their usefulness has been documented in hospitalized patients with COVID-19, complementing established diagnostic techniques [20]. However, little is known about the performance of these tests—particularly quantitative assays—in LTCF populations following COVID-19 vaccination.

Therefore, the aim of this study was to compare several qualitative serological tests for COVID-19 with a quantitative assay, in relation to vaccination status and disease course, using serum samples from elderly LTCF residents. The prospective design was justified by the lack of existing studies in Poland specifically targeting this population. At the time, the COVID-19 situation was highly dynamic and rapidly evolving, further supporting the need for a real-time, prospective approach to capture developments in this vulnerable group.

## 2. Materials and Methods

### 2.1. Residents

Residents were recruited into a non-interventional, prospective, multicenter study aimed at assessing vaccination coverage among 429 individuals living in eight long-term care facilities (LTCFs) in Southern Poland, including both residential and nursing homes. The residents were in varying health conditions; 208 experienced difficulties with mobility and feeding.

Data was collected between January and June 2022 by trained medical staff. Inclusion criteria included the ability to provide informed consent and current residency in an LTCF. Written informed consent was obtained from each participant, confirming their understanding of the study’s aims and procedures. Each participant completed a survey, followed by blood sample collection.

No individuals were excluded from studying. Analysis of the vaccination campaign in LTCFs revealed that 382 participants (89%) had received one or two doses of a COVID-19 vaccine, depending on the vaccine type. Among them, 102 residents received one additional (booster) dose, and 228 received two additional doses.

The questionnaire included sections on sociodemographic data, medical history (including comorbidities), and COVID-19-related information. The history of symptomatic or asymptomatic COVID-19, including disease severity and vaccination status, was determined based on medical records. Participants without confirmed SARS-CoV-2 infection were classified as uninfected.

The study received approval from the Bioethics Committee of Jagiellonian University, under number: 1072.6120.212.2021. Funding was provided by the National Science Centre Poland, grant number: 2020/39/B/NZ6/01939.

### 2.2. Serum Samples

Blood samples were collected from residents between January and December 2022 by professional medical staff, using tubes without anticoagulant. Samples were drawn in the morning, either on an empty stomach or after a light breakfast and were promptly transported to the laboratory of the Chair of Microbiology at the Jagiellonian University Medical College for further processing and analysis.

SARS-CoV-2 infection in patients was confirmed by RT-PCR testing conducted by regional health authorities.

### 2.3. Enzyme-Linked Immunosorbent Assays (ELISA)

Blood samples from patients were collected in tubes without anticoagulants. After collecting, the tubes were allowed to coagulate for 30 to 60 min at room temperature. After this time, the clotted blood was centrifuged for 8 min at 1250 rpm. The serum was then aliquoted into small aliquots and froze at −80 °C until use.

Following tests produced by Euroimmun (Luebeck, Germany) were used: Anti-SARS-CoV-2 NCP ELISA (IgG) for serum, catalogue number: EI 2606-9601-2 G, Anti-SARS-CoV-2 ELISA (IgA); cat. no.: EI 2606-9601 A, Anti-SARS-CoV-2 NCP ELISA (IgM) cat. no: EI 2606-9601-2M and Anti-SARS-CoV-2 (IgG) cat. no: EI 2606-9601 G. The quantitative Anti-SARS-CoV-2 QuantiVac ELISA (IgG) cat. no.: EI 2606-9601-10 G test was also applied and used as a comparator. The absorbance of the samples was measured immediately at 450 nm and a reference wavelength of 630 nm using the microplate reader (BioTek ELx808 and Multiskan FC Thermo Scientific, Vantaa, Finland. The data was then analyzed according to the manufacturer’s instructions. The results are presented in accordance with the following formula: Control or Patient Extinction/Calibrator Extinction = Ratio. The results were interpreted according to the manufacturer’s recommendation based on the Ratio factor. Ratio < 0.8 negative, Ratio ≥0.8 to <1.1 borderline, Ratio ≥ 1.1 positive. The following interpretation of the results was recommended for the quantitative Anti-SARS-CoV-2 QuantiVac ELISA (IgG): lower than 8 RU/mL as negative; between 8 and 11 RU/mL as borderline; and ≥11 RU/mL as positive, where RU/mL is the unit of sample concentration.

### 2.4. Statistical Analysis

In the statistical analysis, relative and absolute frequencies were used for nominal variables, and the mean value with standard deviation was used for quantitative variables. For the non-normal distribution of data, median with interquartile range (IQR) was used. For the group comparison, the U-Mann–Whitney test was used. Correlations were tested including Spearman’s correlation due to the non-normal distribution of the data. The analysis was conducted using the International Business Machines Corporation Statistical Package for the Social Sciences, version 29 (IBM Corporation, Armonk, NY, USA). In all analyses, the significance level was set at α = 0.05.

## 3. Results

We analyzed samples from 429 residents of Polish LTCFs, the majority of whom were women (65%). The mean age was 80 years.

Among the seven different qualitative serological tests selected to detect anti-SARS-CoV-2 antibodies, only the Anti-SARS-CoV-2 NCP (IgG) and Anti-SARS-CoV-2 (IgA) tests showed a positive correlation with the Anti-SARS-CoV-2 QuantiVac (IgG) test, which was used as a comparator. Additionally, the Anti-SARS Coronavirus IIFT (IgG) test demonstrated a positive, though weaker, correlation (Figure 1).

Among the most frequently recorded comorbidities in the studied residents, only degenerative joint disease and diabetes showed a weak correlation with higher anti-SARS-CoV-2 antibody levels, as measured by the Anti-SARS-CoV-2 QuantiVac (IgG) assay (Figure 2).

Among the various data describing the course of COVID-19 and the vaccination status of residents, vaccination was associated with elevated antibody titers, as measured by the Anti-SARS-CoV-2 QuantiVac (IgG) assay. As expected, a strong correlation was observed between positive RT-PCR test results and higher antibody titters detected by the same assay. Notably, a fully symptomatic course of the disease did not show a strong correlation with antibody levels (Figure 3).

Spearman’s correlation analysis was used to compare selected variables with Anti-SARS-CoV-2 QuantiVac IgG antibody concentrations due to the non-normal distribution of the data. A strong positive correlation was observed between Anti-SARS-CoV-2 QuantiVac IgG levels and both Anti-SARS-CoV-2 IgA (rs = 0.553, *p* < 0.001) and Anti-SARS-CoV-2 NCP (IgG) (rs = 0.332, *p* < 0.001). A weak correlation was noted between Anti-SARS-CoV-2 QuantiVac IgG concentrations and the age of the residents. No correlation was found between IgG levels and the number of days since the last exposure to SARS-CoV-2, nor with other demographic variables.

## 4. Discussion

Residents of long-term care facilities (LTCFs) were among the most affected populations during the COVID-19 pandemic, experiencing high morbidity and mortality due to their increased vulnerability to infections [21]. The most significant risk factors for COVID-19-related morbidity include advanced age and communal living conditions, which facilitate the spread of infections—particularly SARS-CoV-2—through various transmission routes in enclosed environments [22].

In a previous study on anti-SARS-CoV-2 antibodies conducted in the same cohort of residents, we observed that dependent individuals exhibited higher antibody levels than their independent counterparts, despite having received the same vaccination regimen. In both that study and the present one, only a weak correlation was found between residents’ age and anti-SARS-CoV-2 antibody levels. This somewhat unexpected result may be attributed to the well-documented phenomenon of age-related chronic inflammation. Xu and Larbi [23], in their review on immunity and inflammation in the elderly, emphasized that while inflammation is essential for immune activation, its dysregulation can lead to chronic inflammation, which may be detrimental to the host. Moreover, age-related polyclonal activation has been associated with reduced antibody affinity, which could explain why higher antibody titers in older individuals may nonetheless exhibit lower functional activity and, consequently, reduced protective efficacy [24].

Numerous serology-based assays for detecting anti-SARS-CoV-2 antibodies were developed and approved for clinical use during the pandemic. These tests aim to assess individual seroprotection status, although their sensitivity and specificity vary [25]. Most commonly, these assays are based on lateral-flow immunoassays (LFIA) or enzyme-linked immunosorbent assays (ELISAs), with no significant differences in performance between platforms [26].

SARS-CoV-2-specific antibodies represent the most effective humoral immune mechanism for reducing viral virulence and promoting population-level immunity [27]. Neutralization assays using live virus particles are considered the gold standard; however, they are time-consuming, require biosafety level 3 laboratories, are difficult to standardize, and are impractical for large-scale implementation [27]. The QuantiVac ELISA assay, which quantitatively measures anti-S1 IgG antibodies, has shown strong correlation with neutralization assays and can be used as a comparator to confirm and monitor recent and past SARS-CoV-2 infections [18,19]. Therefore, we used this assay as a comparator in our study.

Only two ELISA-based serological tests and one IIFT-based test showed positive correlation with the QuantiVac ELISA, suggesting their potential utility for rapid qualitative screening of anti-SARS-CoV-2 antibodies in elderly populations. However, QuantiVac cannot be used to calculate the positive and negative predictive values (PPV and NPV) of other ELISAs due to differences in the immunoglobulin classes detected. As Böger [28] emphasized, although RT-PCR remains the gold standard for diagnosing active COVID-19, it is not suitable for monitoring treatment effectiveness. A combination of clinical, molecular, and serological diagnostic methods is recommended to achieve optimal sensitivity and specificity.

In our study, vaccination status was positively correlated with high antibody titers measured by QuantiVac, which are known to correspond with virus neutralization [18,19]. This finding strongly supports the prevailing view that vaccination is the most effective strategy for protecting elderly LTCF residents against clinical COVID-19. Our results align with the opinion expressed by Xu et al. [29], who advocate for COVID-19 vaccination as a recommended strategy to prevent SARS-CoV-2 infection and related mortality in older adults.

Interestingly, the presence of IgA antibodies against the S1 antigen, detected by the Anti-SARS-CoV-2 (IgA) test, was well correlated with higher IgG titters to the same protein measured by QuantiVac ELISA. IgA antibodies are known to persist in the serum of healthy adults similarly to IgG [30]. Moreover, studies by Gaebler et al. [31] have shown that IgM and IgG titers against the receptor-binding domain (RBD) of the spike protein decline significantly over time, while IgA levels are less affected—a trend that appears to apply to elderly populations as well. The sensitivity of other serological tests, compared to the QuantiVac comparator, was generally lower, except for the Anti-SARS Coronavirus IIFT (IgG), which showed a modest correlation.

We also observed an interesting correlation between higher anti-SARS-CoV-2 antibody titers measured by QuantiVac ELISA and the presence of diabetes or degenerative joint disease in some residents. This may be explained by the general hyperactivity of the immune system in these conditions. For instance, in obesity-associated osteoarthritis, cell B activation leads to increased antibody production [32]. Additionally, COVID-19 has been shown to trigger autoimmune diseases such as psoriasis, rheumatoid arthritis, and diabetes, often accompanied by elevated levels of antinuclear antibodies [33,34].

All serological tests used in this study were selected from a broad panel of assays approved for detecting antibodies against various SARS-CoV-2 antigens and were produced by a single manufacturer to minimize selection bias. Furthermore, there was no recruitment bias, as the study population represented a highly homogeneous group of residents from Polish nursing homes and related institutions. This same population has been studied by our team for several years, including prior to the COVID-19 pandemic [34].

## 5. Limitations

This study has several limitations that should be acknowledged to provide a balanced interpretation of the findings. First, the absence of viral neutralization assays limits our ability to assess the functional capacity of the detected antibodies. While binding antibody levels offer valuable insights into humoral responses, they do not directly reflect neutralizing activity, which is more closely associated with protective immunity.

Second, the retrospective collection of infection history data may be subject to recall bias or inaccuracies, particularly in elderly residents or in cases where documentation was incomplete. This could have influenced the classification of participants’ prior SARS-CoV-2 exposure and, consequently, the interpretation of immune responses.

Third, there was variability in the time intervals between vaccination or infection and the timing of blood sample collection. This heterogeneity may have affected the measured antibody levels due to natural waning over time, potentially introducing bias in the comparison between individuals or subgroups.

## 6. Conclusions

Despite these limitations, the study provides valuable real-world data on post-vaccination humoral responses in a vulnerable population residing in LTCFs and highlights important associations with clinical and demographic factors. Our results confirmed that vaccination positively influences antibody responses, underscoring the importance of immunization among LTCF residents. Additionally, certain comorbidities—such as degenerative joint disease and diabetes—showed weak correlations with higher antibody levels.

## Figures and Tables

**Figure 1 idr-17-00089-f001:**
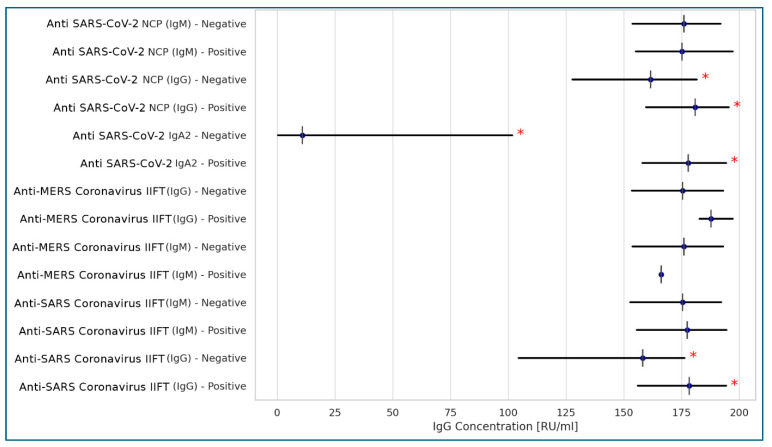
Correlations between Anti-SARS-CoV-2 QuantiVac (IgG) test and 7 qualitative serologic tests for anti-SARS-CoV-2 coronavirus antibodies (Spearman’s correlation). Legend: * statistically significant results at *p* < 0.001; <0.2: no correlation; 0.2–0.39: weak correlation; 0.4–0.59: moderate correlation; >0.6: strong correlation.

**Figure 2 idr-17-00089-f002:**
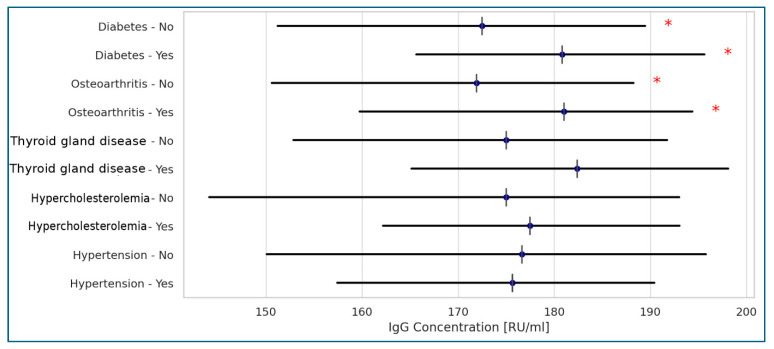
Correlations between Anti-SARS-CoV-2 QuantiVac (IgG) concentrations and diseases present in the tested populations of the Polish LTCF residents (Spearman’s correlation). Legend: * statistically significant results at *p* < 0.001; <0.2: no correlation; 0.2–0.39: weak correlation; 0.4–0.59: moderate correlation; >0.6: strong correlation.

**Figure 3 idr-17-00089-f003:**
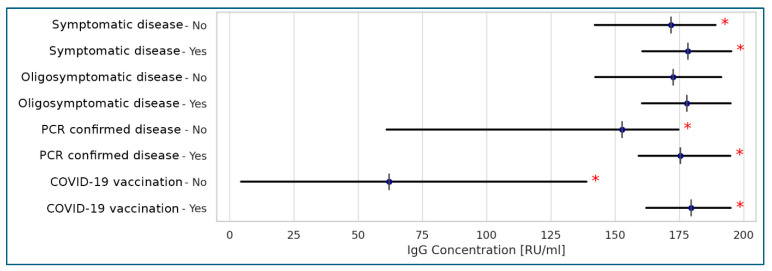
Correlations between anti-SARS-CoV-2 QuantiVac (IgG) concentrations and the disease course and vaccinations in the tested populations of the Polish LTCF residents (Spearman’s correlation). Legend: * statistically significant results at *p* < 0.001; <0.2: no correlation; 0.2–0.39: weak correlation; 0.4–0.59: moderate correlation; >0.6: strong correlation.

## Data Availability

The datasets analyzed during the current study are available from Justyna Brodowicz (e-mail: justyna.brodowicz@uj.edu.pl) upon reasonable request.

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
