# Peer review of "A Multi-Center Prospective Study on Post-Vaccination Humoral Response to SARS-CoV-2 in Polish Long-Term Care Facility Residents: Associations with COVID-19 Clinical Course and Comorbidities"

_2036-7449, 2025, doi:10.3390/idr17040089_

Round 1
Reviewer 1 Report
Comments and Suggestions for Authors
Keywords should be written in scientific language.
Clarity and language: Improve your scientific writing to achieve greater objectivity and cohesion.
Methodology: Justify the choice of a prospective design and provide more detail on the LTCIEs and the sample.
Ethical aspects: Explain how the residents' ability to consent was assessed.
Statistical analyses: Consider including multivariate analyses to control for confounders.
Antibody levels: Discuss whether the titres found are associated with relevant clinical protection.
Influenza/pneumococcal vaccines: Include data on coverage of these vaccines, or justify the absence of such data.
Comorbidities: Explore possible mechanisms for the association between comorbidities and antibodies.
Practical implications: Suggest how the findings could inform vaccination policies in LTCIs.
Limitations: Discuss limitations such as selection bias and the sensitivity of serological tests.
Comments on the Quality of English LanguageThe English could be improved.
Author Response
DETAILED RESPONSE TO REVIEWER, Round 1
STEP-BY-STEP REPLIES TO REVIEWER COMMENTS:
Reviewer 1
Keywords should be written in scientific language.
Authors’ reply: Due to your kind suggestion we made appropriate corrections.
Clarity and language: Improve your scientific writing to achieve greater objectivity and cohesion.
Authors’ reply: As suggested, the English language of the article was corrected.
Methodology: Justify the choice of a prospective design and provide more detail on the LTCIEs and the sample.
Authors’ reply: Due to your kind suggestion we made appropriate corrections: the “aim” and the “Materials and methods” section were supplemented. The choice of a prospective design was justified by the lack of existing studies in Poland specifically targeting patients residing in long-term care facilities. At the time, the COVID-19 pandemic situation was highly dynamic and rapidly evolving, which further supported the need for a prospective approach to capture real-time data and developments within this specific population. We also add more details on the LTCIEs and the sample.
Ethical aspects: Explain how the residents' ability to consent was assessed.
Authors’ reply: Due to your kind suggestion we made appropriate corrections: the “Materials and methods” section was supplemented.
Statistical analyses: Consider including multivariate analyses to control confounders
Authors’ reply: Due to lack of clinical data and other covariates we were unable to build significant multivariate analysis model. In further studies there are planned such analysis due to ability to combine more clinical relevant data
Antibody levels: Discuss whether the titers found are associated with relevant clinical protection.
Authors’ reply: There was indeed a correlation between high titers and vaccination, but post-vaccination period clinical observations were not performed. A sentence on relationship between high titers and vaccination was added.
Influenza/pneumococcal vaccines: Include data on coverage of these vaccines, or justify the absence of such data.
Authors’ reply: The topic of the influenza/pneumococcal vaccine coverage has been addressed in another paper that is currently in peer review: "Are they too old for vaccination? Ageism and disparities in immunization among older residents of long-term care facilities in Southern Poland" submitted to Vaccine Elsevier (under review)
Comorbidities: Explore possible mechanisms for the association between comorbidities and antibodies.
Authors’ reply: No such association was found in this study
Practical implications: Suggest how the findings could inform vaccination policies in LTCIs. Two sentences to Discussion
Authors’ reply: Due to your kind suggestion we made appropriate corrections, the “Discussion” section was supplemented.
Limitations: Discuss limitations such as selection bias and the sensitivity of serological tests.
Authors’ reply: Due to your kind suggestion we made appropriate corrections, the “Discussion” section was supplemented.
Reviewer 2 Report
Comments and Suggestions for Authors
General Comments
This manuscript presents a prospective, observational study evaluating the performance of different SARS-CoV-2 serological tests in elderly residents of long-term care facilities (LTCFs) in southern Poland. The study is timely and relevant, given the ongoing importance of post-vaccination immunity monitoring in vulnerable populations.
However, significant revisions are needed before the article can be considered for publication. These relate primarily to language quality, methodological transparency, clarity of results, and the depth of the discussion.
Major Points
Title: The current title is too generic. It does not specify which vaccine is being evaluated. Since the focus is on COVID-19 vaccination, the title should explicitly reference it. A more precise and informative title would help readers understand the study's scope immediately.
Study Design and Test Selection Justification: The manuscript does not report the number of participants included in the study. This is a major omission. The total sample size, number of excluded individuals (if any), and group sizes for key comparisons (e.g., vaccinated vs. unvaccinated) must be clearly stated in both the Methods and Results sections. Without this information, the interpretation of statistical analyses is severely limited.
Data Presentation: Tables 1–3 are informative but would benefit from visual representation of antibody titers (e.g., boxplots or violin plots) to aid interpretation. Clarify criteria used to define “strong” or “weak” correlations (e.g., Spearman’s rs thresholds).
Discusion: The Discussion section is notably weak and requires substantial revision to meet the standards of a scientific publication. Currently, it largely restates the results without offering sufficient depth of interpretation, integration with existing literature, or exploration of clinical and public health implications.
The authors fail to critically compare their findings with prior studies on post-COVID-19 vaccination immune responses in elderly or institutionalized populations, missing the opportunity to position their results within the broader scientific context.
While the association between higher antibody levels and comorbidities such as diabetes or joint disease is interesting, the mechanistic explanation remains speculative. This part of the discussion should be rewritten with greater caution and supported by clearer referencing to existing immunological evidence.
Moreover, the limitations of the study are currently underdeveloped and should be more clearly acknowledged to provide a balanced interpretation of the findings. Specifically, the authors are encouraged to address the absence of viral neutralization tests, which limits the assessment of functional antibody responses. Additionally, potential recall bias or inaccuracies in historical data (particularly regarding infection history) should be considered. The authors should also note the variability in the time intervals between vaccination or infection and blood sampling, as this may affect antibody kinetics.
Finally, it is important to recognize the possible influence of unaccounted confounding factors that could have impacted the observed associations. A thorough reworking of the Discussion section is essential to enhance the manuscript’s clarity, scientific rigor, and interpretative value.
- Ethics and Consent
Although ethics approval is stated, please clarify whether written informed consent was obtained from participants or their legal representatives, considering their vulnerable status.
Minor Comments
- Abstract: Consider specifying the total number of participants in the abstract.
- Introduction: A more concise explanation of viral structure and immunogenic regions would improve focus.
- Reference 9 (“Bylica et al., in press”) (Line 197) should be updated in the main text
- Typographical inconsistencies in tables (e.g., misaligned columns) should be corrected.
- The manuscript requires comprehensive language editing by a native English speaker or professional editor. Numerous grammatical, syntactic, and stylistic errors affect readability and professional tone Examples include: “This elements/factors characteristic…” should read “These characteristics…”redundant phrasing, such as “fully symptomatic disease course was not fully correlated…”
Conclusion
This study has the potential to provide valuable insight into SARS-CoV-2 immunity monitoring among institutionalized older adults. However, substantial revisions are needed to address methodological clarity, improve the writing, and appropriately contextualize the findings. I recommend major revision before further consideration.
Author Response
DETAILED RESPONSE TO REVIEWER, Round 1
STEP-BY-STEP REPLIES TO REVIEWER COMMENTS:
Reviewer 2
General Comments
This manuscript presents a prospective, observational study evaluating the performance of different SARS-CoV-2 serological tests in elderly residents of long-term care facilities (LTCFs) in southern Poland. The study is timely and relevant, given the ongoing importance of post-vaccination immunity monitoring in vulnerable populations.
Authors’ reply: Thank you for this comment!
However, significant revisions are needed before the article can be considered for publication. These relate primarily to language quality, methodological transparency, clarity of results, and the depth of the discussion.
Major Points
- Title: The current title is too generic. It does not specify which vaccine is being evaluated. Since the focus is on COVID-19 vaccination, the title should explicitly reference it. A more precise and informative title would help readers understand the study's scope immediately.
Authors’ reply: Due to your kind suggestion we made appropriate corrections, the the current title is: “Post-Vaccination Humoral Response to SARS-CoV-2 in Polish Long-Term Care Facility Residents: Associations with COVID-19 Clinical Course and Comorbidities: Multi-Center Prospective Study”
- Study Design and Test Selection Justification: The manuscript does not report the number of participants included in the study. This is a major omission. The total sample size, number of excluded individuals (if any), and group sizes for key comparisons (e.g., vaccinated vs. unvaccinated) must be clearly stated in both the Methods and Results sections. Without this information, the interpretation of statistical analyses is severely limited.
Authors’ reply: Due to your kind suggestion we made appropriate corrections, both “Abstract” and the “Material and Methods” section were supplemented.
- Data Presentation: Tables 1–3 are informative but would benefit from visual representation of antibody titers (e.g., boxplots or violin plots) to aid interpretation. Clarify criteria used to define “strong” or “weak” correlations (e.g., Spearman’s rs thresholds).
Authors’ reply: Due to your kind suggestion we made appropriate corrections; to complement the tabular data, visual representations of antibody titters are provided in Figures 1–3, allowing for a clearer comparison of distributions across study groups. Spearman correlation was assessed for criteria used in medical research in the following ranges: <0.2 no correlation, 0.2-0.39 weak correlation, 0.4-0.59 moderate correlation, >0.6 strong correlation
- Discusion: The Discussion section is notably weak and requires substantial revision to meet the standards of a scientific publication. Currently, it largely restates the results without offering sufficient depth of interpretation, integration with existing literature, or exploration of clinical and public health implications.
Authors’ reply: Due to your kind suggestion we made appropriate corrections, the “Discussion” section was supplemented.
- The authors fail to critically compare their findings with prior studies on post-COVID-19 vaccination immune responses in elderly or institutionalized populations, missing the opportunity to position their results within the broader scientific context.
Authors’ reply: We have corrected the "discussion" as indicated in the point above, now nearly all positions of the existing literature on Covid-19 and immune response to it in elderly was presented and discussed.
- While the association between higher antibody levels and comorbidities such as diabetes or joint disease is interesting, the mechanistic explanation remains speculative. This part of the discussion should be rewritten with greater caution and supported by clearer referencing to existing immunological evidence.
Authors’ reply: In fact, there were no correlations between co-morbidities and antibody titers and therefore, we have not discussed these associations in “Discussion” section.
- Moreover, the limitations of the study are currently underdeveloped and should be more clearly acknowledged to provide a balanced interpretation of the findings. Specifically, the authors are encouraged to address the absence of viral neutralization tests, which limits the assessment of functional antibody responses.
Authors’ reply: Due to your kind suggestion we made appropriate corrections, the “Limitations” section was supplemented, in fact, as documented in the literature, anti-SARS-CoV-2 QuantiVac (IgG) test results are fully correlated with results of the viral neutralization tests. After discussion with our infectiologists, we decided to omit the neutralization test.
- Additionally, potential recall bias or inaccuracies in historical data (particularly regarding infection history) should be considered. The authors should also note the variability in the time intervals between vaccination or infection and blood sampling, as this may affect antibody kinetics.
Authors’ reply: Due to your kind suggestion we made appropriate corrections, an appropriate paragraph on bias was added. Since the persistence of the anti-SARS-Cov-2 antibodies after infection or vaccination is calculated for about 6 months, antibody kinetic study was not performed since no one participant showed longer period between vaccination and antibody testing.
- Finally, it is important to recognize the possible influence of unaccounted confounding factors that could have impacted the observed associations. A thorough reworking of the Discussion section is essential to enhance the manuscript’s clarity, scientific rigor, and interpretative value.
Authors’ reply: Due to your kind suggestion we made appropriate corrections, the “Discussion” section was re-writing, as described above.
- Ethics and Consent: Although ethics approval is stated, please clarify whether written informed consent was obtained from participants or their legal representatives, considering their vulnerable status.
Authors’ reply: Due to your kind suggestion we made appropriate corrections: the “Materials and methods” section was supplemented.
Minor Comments
- Abstract: Consider specifying the total number of participants in the abstract.
Authors’ reply: Corrected according to suggestion.
- Introduction: A more concise explanation of viral structure and immunogenic regions would improve focus.
Authors’ reply: Supplemented and corrected according to suggestion.
- Reference 9 (“Bylica et al., in press”) (Line 197) should be updated in the main text
Authors’ reply: This paper is still unpublished.
- Typographical inconsistencies in tables (e.g., misaligned columns) should be corrected.
Authors’ reply: Corrected according to suggestion.
- The manuscript requires comprehensive language editing by a native English speaker or professional editor. Numerous grammatical, syntactic, and stylistic errors affect readability and professional tone Examples include: “This elements/factors characteristic…” should read “These characteristics…” redundant phrasing, such as “fully symptomatic disease course was not fully correlated…”
Authors’ reply: As suggested, the English language of the article was corrected..
Conclusion
This study has the potential to provide valuable insight into SARS-CoV-2 immunity monitoring among institutionalized older adults. However, substantial revisions are needed to address methodological clarity, improve the writing, and appropriately contextualize the findings. I recommend major revision before further consideration.
Authors’ reply: Thank you for this comment!
Reviewer 3 Report
Comments and Suggestions for Authors
This serological study essentially compares the performance of a number of “rapid” qualitative SARS-CoV-2 antibody tests with a well-established QuantiVac®, which is a quantitative ELISA that has been show to correlate well with the gold standard neutralization test, in a Polish population of long-term care facility residents. In addition the correlation of the QuantiVac test with COVID disease presentation and vaccination status is also investigated. The weak positive correlation of rapid tests, based on IgG, with QuantiVac was expected, but the very strong correlation of an IgA-based rapid test with QuantiVac is remarkable. As could be expected, there is a strong correlation of QuantiVac with RT-PCR confirmed SARS-CoV-2 infection and with COVID vaccination, but a much weaker correlations with (previous) COVID disease, diabetes and degenerative joint disease, the latter probably representing conditions with polyclonal B cell activation.
The title is not representative of the content, since vaccination is not the only focus. It should be changed therefore. The findings are not groundbreaking, they add some specific data to a large body of already existing knowledge, but fail to fully analyze and compare them with the literature. The statistical analysis is limited to Mann-Whitney U and Spearman’s rank correlation tests. The question of clinical significance however is to what extent the rapid tests are reliable diagnostics in this population. To that end, the positive and negative predictive value of each rapid test as compared to the QuantiVac (considered as gold standard) should be calculated and presented. In order to provide insightful information, these date on predictive value should then be put into perspective of what the manufacturers claim and what has been published previously by independent investigators.
Author Response
DETAILED RESPONSE TO REVIEWER, Round 1
STEP-BY-STEP REPLIES TO REVIEWER COMMENTS:
Reviewer 3
This serological study essentially compares the performance of a number of “rapid” qualitative SARS-CoV-2 antibody tests with a well-established QuantiVac®, which is a quantitative ELISA that has been show to correlate well with the gold standard neutralization test, in a Polish population of long-term care facility residents. In addition, the correlation of the QuantiVac test with COVID disease presentation and vaccination status is also investigated. The weak positive correlation of rapid tests, based on IgG, with QuantiVac was expected, but the very strong correlation of an IgA-based rapid test with QuantiVac is remarkable. As could be expected, there is a strong correlation of QuantiVac with RT-PCR confirmed SARS-CoV-2 infection and with COVID vaccination, but a much weaker correlations with (previous) COVID disease, diabetes and degenerative joint disease, the latter probably representing conditions with polyclonal B cell activation.
Authors’ reply: Thank you for this comment! We have added a sentence on persistence of IgA anti-SARS-CoV-2 antibodies to “Discussion “ section.
The title is not representative of the content, since vaccination is not the only focus. It should be changed therefore.
Authors’ reply: Due to your kind suggestion we made appropriate corrections, the the current title is: “Post-Vaccination Humoral Response to SARS-CoV-2 in Polish Long-Term Care Facility Residents: Associations with COVID-19 Clinical Course and Comorbidities: Multi-Center Prospective Study”
The findings are not groundbreaking, they add some specific data to a large body of already existing knowledge, but fail to fully analyze and compare them with the literature. The statistical analysis is limited to Mann-Whitney U and Spearman’s rank correlation tests. The question of clinical significance however is to what extent the rapid tests are reliable diagnostics in this population. To that end, the positive and negative predictive value of each rapid test as compared to the QuantiVac (considered as gold standard) should be calculated and presented. In order to provide insightful information, these date on predictive value should then be put into perspective of what the manufacturers claim and what has been published previously by independent investigators.
Authors’ reply: Due to your kind suggestion we made appropriate corrections. The ELISAs used in the study are not rapid tests such as immunochromatography and similar. We used the Anti-SARS-COV-2 QuantiVac ELISA as a comparator (unfortunately referred to in the text as the "gold standard" - corrected) because it was the only quantitative method. However, the QuantiVac cannot be used to calculate the PPV and NPV of the other ELISAs due to the different immunoglobulin classes detected by the individual tests.
Round 2
Reviewer 2 Report
Comments and Suggestions for Authors
The manuscript was thoroughly revised and corrected based on my suggestions
Author Response
The manuscript was thoroughly revised and corrected based on my suggestions
Authors’ reply: Thank you for this comment!
Reviewer 3 Report
Comments and Suggestions for Authors
The authors have significantly improved the clarity of all the sections in their manuscript. They have correctly adapted the title and appropriately explained why calculation of positive and negative predictive value was not performed.
The Figures and Tables provide exactly the same information, but the figures are much clearer. Therefore the tables should be omitted.
The weakly positive correlation of anti-SARS-CoV-2 antibodies with age is at first view indeed somewhat paradoxical, in view of the increased susceptibility to disease with high age. This phenomenon could be related to inflammageing, as proposed in the second paragraph of the Discussion. The authors should also mention that age-related polyclonal activation has also been associated with decreased antibody affinity, which then could explain why these higher antibody titers nevertheless have lower functional activity, hence are less protective; See https://pmc.ncbi.nlm.nih.gov/articles/PMC8524000/pdf/fimmu-12-733566.pdf
Author Response
The authors have significantly improved the clarity of all the sections in their manuscript. They have correctly adapted the title and appropriately explained why calculation of positive and negative predictive value was not performed.
Authors’ reply: Thank you for this comment!
The Figures and Tables provide exactly the same information, but the figures are much clearer. Therefore the tables should be omitted.
Authors’ reply: Corrected according to suggestions, tables were removed.
The weakly positive correlation of anti-SARS-CoV-2 antibodies with age is at first view indeed somewhat paradoxical, in view of the increased susceptibility to disease with high age. This phenomenon could be related to inflammageing, as proposed in the second paragraph of the Discussion. The authors should also mention that age-related polyclonal activation has also been associated with decreased antibody affinity, which then could explain why these higher antibody titers nevertheless have lower functional activity, hence are less protective; See https://pmc.ncbi.nlm.nih.gov/articles/PMC8524000/pdf/fimmu-12-733566.pdf
Authors’ reply: Corrected according to suggestions, the “Discussion” section was supplemented, as below:
“(…) In a previous study on anti-SARS-CoV-2 antibodies conducted in the same cohort of residents [Bylica et al., in press], we observed that dependent individuals exhibited higher antibody levels than their independent counterparts, despite having received the same vaccination regimen. In both that study and the present one, only a weak correlation was found between residents’ age and anti-SARS-CoV-2 antibody levels. This somewhat unexpected result may be attributed to the well-documented phenomenon of age-related chronic inflammation. Xu and Larbi [23], in their review on immunity and inflammation in the elderly, emphasized that while inflammation is essential for immune activation, its dysregulation can lead to chronic inflammation, which may be detrimental to the host. Moreover, age-related polyclonal activation has been associated with reduced antibody affinity, which could explain why higher antibody titers in older individuals may nonetheless exhibit lower functional activity and, consequently, reduced protective efficacy [24]. (…)”